# NEURAL POOLING FOR GRAPH NEURAL NETWORKS

## ABSTRACT

Tasks such as graph classification, require graph pooling to learn graph-level representations from constituent node representations. In this work, we propose two novel methods using fully connected neural network layers for graph pooling, namely Neural Pooling Method 1 and 2. Our proposed methods have the ability to handle variable number of nodes in different graphs, and are also invariant to the isomorphic structures of graphs. In addition, compared to existing graph pooling methods, our proposed methods are able to capture information from all nodes, collect second-order statistics, and leverage the ability of neural networks to learn relationships among node representations, making them more powerful. We perform experiments on graph classification tasks in the bio-informatics and social network domains to determine the effectiveness of our proposed methods. Experimental results show that our methods lead to an absolute increase of upto 1.2% in classification accuracy over previous works and a general decrease in standard deviation across multiple runs indicating greater reliability. Experimental results also indicate that this improvement in performance is consistent across several datasets.

## 1 INTRODUCTION

Over the past several years, there is a growing number of applications where data is generated from non-Euclidean domains and is represented as graphs with complex relationships and interdependency between entities. Deep learning generalised from grid-like data to the graph domain has led to the development of the remarkably successful Graph Neural Networks (GNNs) (Fan et al., 2019; Gao et al., 2019; Ma et al., 2019a; Wang et al., 2019b) and its numerous variants such the Graph Convolutional Network (GCN) (Kipf & Welling, 2017), GraphSAGE (Hamilton et al., 2017), graph attention network (GAT) (Veličković et al., 2018), jumping knowledge network (JK) (Xu et al., 2018), and graph isomorphism networks (GINs) (Xu et al., 2019), etc.

Pooling is a common operation in deep learning on grid-like data, such as images. Pooling layers provide an approach to down sampling feature maps by summarizing the presence of features in patches of the feature map. It reduces dimensionality and also provides local translational invariance. In the case of graph data, pooling is used to obtain a representation of a graph using its constituent node representations. However, it is challenging to develop graph pooling methods due to the some special properties of graph data such as the variable number of nodes in different graphs and the isomorphic structures of graphs. Firstly, the number of nodes varies in different graphs, while the graph representations are usually required to have the same fixed size to fit into other downstream machine learning models where they are used for tasks such as classification. Therefore, graph pooling should be capable of handling the variable number of node representations as inputs and producing fixed-sized graph representations. Secondly, unlike images and texts where we can order pixels and words according to the spatial structural information, there is no inherent ordering relationship among nodes in graphs. Therefore, isomorphic graphs should have the same graph representation, and hence, graph pooling should give the same output by taking node representations in any order as inputs.

Our main contributions in this work are two novel graph pooling methods, Neural Pooling Method 1 and 2. These new pooling methods allow us to do the following,i) produce the same dimensional graph representation for graphs with variable number of nodes, ii) remain invariant to the isomorphic structures of graphs, iii) collect second- order statistics, iv) leverage trainable parameters in the form

of fully connected neural networks to learn relationships between underlying node representations to generate high quality graph representations which are then used for graph classification tasks.

Experiments are performed on four benchmark bio-informatics datasets and five popular social network datasets to demonstrate the effectiveness and superiority of our proposed graph pooling methods. Experimental results show that our methods lead to an improvement in classification accuracy over existing methods and are also more reliable as compared to previous works.

## 2 RELATED WORK

### 2.1 GRAPH NEURAL NETWORKS

A graph can be represented by its adjacency matrix and node features. Formally, for a graph $\mathcal{G}$ consisting of $n$ nodes, its topology information can be represented by an adjacency matrix $\boldsymbol{A} \in \{0, 1\}^{n \times n}$ and the node features can be represented as $\boldsymbol{X} \in \mathbb{R}^{n \times d}$, assuming each node has a d-dimensional feature vector. GNNs learn feature representations for different nodes using these matrices (Gilmer et al., 2017). Several approaches are proposed to investigate deep GNNs, and they generally follow a neighborhood information aggregation scheme (Gilmer et al., 2017; Xu et al., 2019; Kipf & Welling, 2017; Hamilton et al., 2017; Veličković et al., 2018). In each step, the representation of a node is updated by aggregating the representations of its neighbors. Graph Convolutional Networks (GCNs) are popular variants of GNNs and inspired by the first order graph Laplacian methods (Kipf & Welling, 2017). Graph pooling is used to connect embedded graphs outputted by GNN layers with classifiers for graph classification. Given a graph, GNN layers produce node representations, where each node is embedded as a vector. Graph pooling is applied after GNN layers to process node representations into a single feature vector as the graph representation. A classifier takes the graph representation and performs graph classification.

### 2.2 GRAPH POOLING

Early studies employ simple methods such as averaging and summation as graph pooling (Xu et al., 2019; Duvenaud et al., 2015; Defferrard et al., 2016). However, averaging and summation do not capture the feature correlation information, curtailing the overall model performance (Zhang et al., 2018). Other studies have proposed advanced graph pooling methods, including DIFFPOOL (Ying et al., 2018), SORT-POOL (Zhang et al., 2018), TOPKPOOL (Gao & Ji, 2019), SAGPOOL (Lee et al., 2019), and EIGENPOOL (Ma et al., 2019b), and achieve great performance on multiple benchmark datasets. EIGENPOOL involves the computation of eigenvectors, which is slow and expensive. DIFFPOOL (Ying et al., 2018) treats the graph pooling as a node clustering problem. A cluster of nodes from the original graph are merged to form a new node in the new graph. DIFFPOOL (Ying et al., 2018) proposes to perform the graph convolution operation on node features to obtain node clustering assignment matrix. Intuitively, the class assignment of a given node should depend on the class assignments of other neighbouring nodes. However, DIFFPOOL does not explicitly consider high-order structural relationships, which we that are important for graph pooling. SORTPOOL (Zhang et al., 2018), TOPKPOOL (Gao & Ji, 2019), and SAGPOOL (Lee et al., 2019) learn to select important nodes from the original graph and use these nodes to build a new graph. They share the similar idea to learn a sorting vector based on node representations, which indicates the importance of different nodes. Then only the top k important nodes are selected to form a new graph while the other nodes are ignored. However, the ignored nodes may contain important features and this information is lost during pooling. It is worth noting that all the graph pooling methods mentioned till now only collect first-order statistics (Boureau et al., 2010). A recent study has proposed second order graph pooling methods $SOPool_{bimap}$ and $SOPool_{attention}$ (Wang & Ji, 2020). In this work, we propose two novel methods using fully connected neural network layers for graph pooling, namely Neural Pooling Method 1 and 2. Compared to existing graph pooling methods, our proposed methods are able to capture information from all nodes, collect second-order statistics, and leverage the ability of neural networks to learn relationships among node representations, making them more powerful.

## 3 METHODOLOGY

### 3.1 PROPERTIES OF GRAPH POOLING

Consider a graph $\mathcal{G} = (\boldsymbol{A}, \boldsymbol{X})$ represented by its adjacency matrix $\boldsymbol{A} \in \{0, 1\}^{n \times n}$ and node feature matrix $\boldsymbol{X} \in \mathbb{R}^{n \times d}$, where $n$ is the number of nodes in $\mathcal{G}$ and $d$ is the dimension of node features. The node features may come from node labels or node degrees. Graph neural networks are known to be powerful in learning good node representation matrix $\boldsymbol{H}$ from $\boldsymbol{A}$ and $\boldsymbol{X}$:

$$\boldsymbol{H} = [\boldsymbol{h}_1, \boldsymbol{h}_2, .........., \boldsymbol{h}_n]^T = \text{GNN}(\boldsymbol{A}, \boldsymbol{X}) \in \mathbb{R}^{n \times f} \tag{1}$$

where rows of $\boldsymbol{H}$, $\boldsymbol{h}_i \in \mathbb{R}^f$, $i = 1, 2, ..., n$, are representations of $n$ nodes, and $f$ is the dimension of the node representation obtained from the GNN and depends on the architecture of the GNN. The task that we focus on in this work is to obtain a graph representation vector $\boldsymbol{h}_G$ from $\boldsymbol{H}$, which is then fed into a classifier to perform graph classification:

$$\boldsymbol{h}_G = \text{g}([\boldsymbol{A}], \boldsymbol{H}) \in \mathbb{R}^c \tag{2}$$

where g($\cdot$) is the graph pooling function and $c$ is the dimension of $\boldsymbol{h}_G$. Here, $[\boldsymbol{A}]$ means that the information from $\boldsymbol{A}$ can be optionally used in graph pooling. For simplicity, we omit it in the following discussion.

Note that the function g($\cdot$) must satisfy two requirements to serve as graph pooling. First, g($\cdot$) should be able to take $\boldsymbol{H}$ with variable number of rows as the inputs and produce fixed-sized outputs. Specifically, different graphs may have different number of nodes, which means that $n$ is a variable. On the other hand, $c$, which is the dimension of the graph representation $\boldsymbol{h}_G$ is supposed to be fixed to fit into the classifier. Second, g($\cdot$) should output the same $\boldsymbol{h}_G$ when the order of rows of $\boldsymbol{H}$ changes. This permutation invariance property is necessary to handle isomorphic graphs. To be concrete, if two graph $\mathcal{G}_1 = (\boldsymbol{A}_1, \boldsymbol{X}_1)$ and $\mathcal{G}_2 = (\boldsymbol{A}_2, \boldsymbol{X}_2)$ are isomorphic, GNNs will output the same multi set of node representations. That is, there exists a permutation matrix $\boldsymbol{P} \in \{0, 1\}^{n \times n}$ such that $\boldsymbol{H}_1 = \boldsymbol{P}\boldsymbol{H}_2$, for $\boldsymbol{H}_1 = \text{GNN}(\boldsymbol{A}_1, \boldsymbol{X}_1)$ and $\boldsymbol{H}_2 = \text{GNN}(\boldsymbol{A}_2, \boldsymbol{X}_2)$. However, the graph representation computed by g($\cdot$) should be the same, i.e., g($\boldsymbol{H}_1$) = g($\boldsymbol{H}_2$) if $\boldsymbol{H}_1 = \boldsymbol{P}\boldsymbol{H}_2$.

### 3.2 NEURAL POOLING METHOD 1

Our first proposed method is called Neural Pooling Method 1. Consider a node representation matrix $\boldsymbol{H}$ obtained following Equation 1 in Section 3.1.

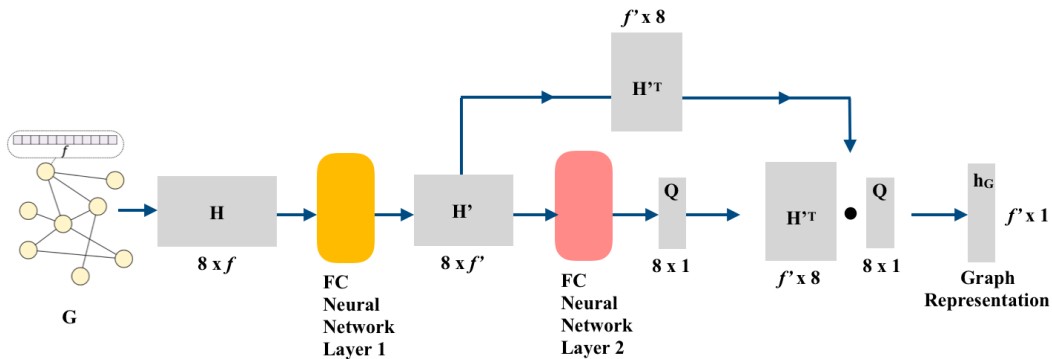

Figure 1: Illustration of our proposed Neural Pooling Method 1. This is an example for a graph $\mathcal{G}$ with 8 nodes. GNNs can learn representations for each node and graph pooling processes node representations into a graph representation vector.

$\boldsymbol{H}$ is passed through a Fully Connected Neural Network Layer ($FCL^1$) to obtain $\boldsymbol{H}'$ as:

$$\boldsymbol{H}' = FCL^1(\boldsymbol{H}) \in \mathbb{R}^{n \times f'} \text{ where } f' < f \tag{4}$$

After this $\boldsymbol{H}'$ is again passed through a second Fully Connected Neural Network Layer ($FCL^2$) to obtain $\boldsymbol{Q}$ as:

$$\boldsymbol{Q} = FCL^2(\boldsymbol{H}') \in \mathbb{R}^{n \times 1} \tag{5}$$

Finally the graph representation $\boldsymbol{h}_G$ is obtained as:

$$\boldsymbol{h}_G = \boldsymbol{H}'^T\boldsymbol{Q} \in \mathbb{R}^{f' \times 1} \tag{6}$$

where $\boldsymbol{H}'^T$ denotes the transpose of $\boldsymbol{H}'$.

Neural Pooling Method 1 always outputs an $f'$-dimensional graph representation for $\boldsymbol{H} \in \mathbb{R}^{n \times f}$, regardless of the value of n. It is also invariant to permutation so that it outputs the same graph representation, even when the order of rows of $\boldsymbol{H}$ changes.

**Intuition**: The $FCL^1$ performs the role of reducing the dimensionality of the input node representations. The trainable parameters of this $FCL^1$ can be thought of as learning a mapping from the $f$ to the $f'$-dimensional space. The $FCL^2$ reduces the $f'$-dimensional node representations to a 1 dimensional representation, $\boldsymbol{Q}$. $\boldsymbol{H}' \in \mathbb{R}^{n \times f'}$ can be viewed as $\boldsymbol{H}' = [\boldsymbol{l}_1, \boldsymbol{l}_2, \ldots, \boldsymbol{l}_{f'}]$, where $\boldsymbol{l}_j \in \mathbb{R}^n$, $j=1, 2, ..., f'$. The vector $\boldsymbol{l}_j$ encodes the spatial distribution of the $j$-th feature in the graph. Based on this view, $\boldsymbol{H}'^T\boldsymbol{Q}$ is able to capture the topology information and $\boldsymbol{Q}$ can be thought of as roughly encoding the position of nodes by learning the weights according to which the $j$-th feature is aggregated across the nodes.

Neural Pooling Method 1 hence, leverages the ability of neural networks to learn the topological structure as well as correlation among the node representations in $\boldsymbol{H}$. It captures the essential features and connections between underlying data. It also reduces the dimensionality of $\boldsymbol{H}$, and results in an accurate representation of the input graph.

## 3.3  NEURAL POOLING METHOD 2

Our second proposed method is called Neural Pooling Method 2. Consider a node representation matrix $\boldsymbol{H}$ obtained following Equation 1 in Section 3.1.

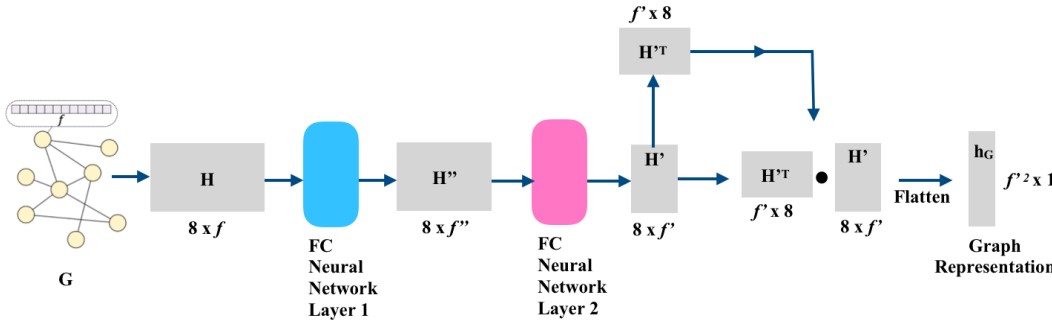

Figure 2: Illustration of our proposed Neural Pooling Method 2. This is an example for a graph $\mathcal{G}$ with 8 nodes. GNNs can learn representations for each node and graph pooling processes node representations into a graph representation vector.

$\boldsymbol{H}$ is passed through a Fully Connected Neural Network Layer ($FCL^1$) to obtain $\boldsymbol{H}''$ as:

$$\boldsymbol{H}'' = FCL^1(\boldsymbol{H}) \in \mathbb{R}^{n \times f''} \text{ where } f'' < f \tag{8}$$

After this $\boldsymbol{H}''$ is again passed through a second Fully Connected Neural Network Layer ($FCL^2$) to obtain $\boldsymbol{H}'$ as:

$$\boldsymbol{H}' = FCL^2(\boldsymbol{H}'') \in \mathbb{R}^{n \times f'} \text{ where } f' < f'' \tag{9}$$

Finally the graph representation $\boldsymbol{h}_G$ is obtained as:

$$\boldsymbol{h}_G = \text{Flatten}(\boldsymbol{H}'^T\boldsymbol{H}') \in \mathbb{R}^{f'^2 \times 1} \tag{10}$$

where $\boldsymbol{H}'^T$ denotes the transpose of $\boldsymbol{H}'$.

**Intuition**: The $FCL^1$ performs the role of reducing the dimensionality of the input node representations. The trainable parameters of this $FCL^1$ can be thought of as learning a mapping from the $f$ to the $f''$-dimensional space. The $FCL^2$ further reduces the $f''$-dimensional node representations to a $f'$ dimensional representation. $\boldsymbol{H}' \in \mathbb{R}^{n \times f'}$ can be viewed as $\boldsymbol{H}' = [\boldsymbol{l}_1, \boldsymbol{l}_2, \ldots, \boldsymbol{l}_{f'}]$, where $\boldsymbol{l}_j \in \mathbb{R}^n$, $j$=1, 2, ..., $f'$. The vector $\boldsymbol{l}_j$ encodes the spatial distribution of the $j$-th feature in the graph. Based on this view, $\boldsymbol{H}'^T \boldsymbol{H}'$ is able to capture the topology information.

Similar to the previous method, Neural Pooling Method 2 satisfies both the conditions of graph pooling which is that it always outputs an $f'^2$-dimensional graph representation for $\boldsymbol{H} \in \mathbb{R}^{n \times f}$, regardless of the value of $n$. It is also invariant to permutation so that it outputs the same graph representation when the order of rows of $\boldsymbol{H}$ changes.

# 4    EXPERIMENTAL SETUP

We perform experiments on graph classification tasks in the bio-informatics and social network domains to demonstrate the effectiveness and superiority of our proposed methods, namely Neural Pooling Methods 1 and 2. Details of datasets and parameter settings are described below.

## 4.1    DATASETS

We use nine graph classification datasets from (Yanardag & Vishwanathan, 2015), including four bioinformatics datasets and five social network datasets. Only bioinformatics datasets come with node labels. For the social network datasets, we use one-hot encoding of node degrees as features. The details of the datasets are summarized in Table 1 and Table 2.

- MUTAG (Debnath et al., 1991) is a bioinformatics dataset of 188 graphs representing nitro compounds. The task is to classify each graph by determining whether the compound is mutagenic aromatic or heteroaromatic.

- PTC (Toivonen et al., 2003) is a bioinformatics dataset of 344 graphs representing chemical compounds. Each node comes with one of 19 discrete node labels. The task is to predict the rodent carcinogenicity for each graph.

- PROTEINS (Borgwardt et al., 2005) is a bioinformatics dataset of 1,113 graph structures of proteins. Nodes in the graphs refer to secondary structure elements (SSEs) and have discrete node labels indicating whether they represent a helix, sheet or turn. And edges mean that two nodes are neighbors along the amino-acid sequence or in space. The task is to predict the protein function for each graph.

- NCI1 (Wale et al., 2008) is a bioinformatics dataset of 4,110 graphs representing chemical compounds. The graph classification label is decided by anti-cancer screens for ability to suppress or inhibit the growth of a panel of human tumor cell lines.

- COLLAB is a scientific collaboration dataset of 5,000 graphs corresponding to ego-networks.The dataset is derived from 3 public collaboration datasets (Leskovec et al., 2005). Each ego-network contains different researchers from each field and is labeled by the corresponding field. The three fields are High Energy Physics, Condensed Matter Physics, and Astro Physics.

- IMDB-BINARY is a movie collaboration dataset of 1,000 graphs representing ego-networks for actors/actresses. The dataset is derived from collaboration graphs on Action and Romance genres. In each graph, nodes represent actors/actresses and edges simply mean they collaborate the same movie. The graphs are labeled by the corresponding genre and the task is to identify the genre for each graph.

- IMDB-MULTI is multi-class version of IMDB-BINARY. It contains 1,500 ego-networks and has three extra genres, namely, Comedy, Romance and Sci-Fi.

- REDDIT-BINARY is a dataset of 2,000 graphs where each graph represents an online discussion thread. Nodes in a graph correspond to users appearing in the corresponding discussion thread and an edge means that one user responded to another. TrollXChromosomes and atheism are discussion-based subreddits, forming two classes to be classified.

Table 1: Details of bioinformatics datasets

| Name | MUTAG | PTC | PROTEINS | NCI1 |
|---|---|---|---|---|
| # graphs | 188 | 344 | 1113 | 4110 |
| # classes | 2 | 2 | 2 | 2 |
| # nodes(max) | 28 | 109 | 620 | 111 |
| # nodes(avg.) | 18.0 | 25.6 | 39.1 | 29.9 |

Table 2: Details of social network datasets

| Name | COLLAB | IMDB-B | IMDB-M | RDT-B | RDT-M5K |
|---|---|---|---|---|---|
| # graphs | 5000 | 1000 | 1500 | 2000 | 5000 |
| # classes | 3 | 2 | 3 | 2 | 5 |
| # nodes(max) | 492 | 136 | 89 | 3783 | 3783 |
| # nodes(avg.) | 74.5 | 19.8 | 13.0 | 429.6 | 508.5 |

- REDDIT-MULTI5K is a similar dataset as REDDIT- BINARY, which contains 5,000 graphs. The difference lies in that REDDIT-MULTI5K crawled data from five different subreddits, namely, worldnews, videos, AdviceAnimals, aww and mildlyinteresting. And the task is to identify the subreddit of each graph instead of determining the type of subreddits.

## 4.2 TRAINING AND EVALUATION

Following (Yanardag & Vishwanathan, 2015; Niepert et al., 2016), model performance is evaluated using 10-fold cross-validation and reported as the average and standard deviation of validation accuracies across the 10 folds. For GNNs, we follow the same training process in (Xu et al., 2019). The GNN has 5 layers. Each multi-layer perceptron (MLP) has 2 layers with batch normalization (Ioffe & Szegedy, 2015). Dropout (Srivastava et al., 2014) is applied in the classifiers. The Adam (Kingma & Ba, 2015) optimizer is used with the learning rate initialized as 0.01 and decayed by 0.5 every 50 epochs. The number of total epochs is selected according to the best cross-validation accuracy. We tune the number of hidden units (16, 32, 64) and the batch size (32, 128) using grid search.

## 4.3 BASELINES

We compare our methods with various other graph pooling methods on the graph classification task, including DIFFPOOL (Ying et al., 2018), SORT-POOL (Zhang et al., 2018), TOPKPOOL (Gao & Ji, 2019), SAGPOOL (Lee et al., 2019), and EIGEN-POOL (Ma et al., 2019b). DIFFPOOL maps nodes to a pre-defined number of clusters but is hard to train. EIGENPOOL involves the computation of eigenvectors, which is slow and expensive. SORTPOOL, SAGPOOL and TOPKPOOL rely on the top-K sorting to select a fixed number (K) of nodes and order them, during which the information from unselected nodes is discarded. We also compare with some recent methods including COVPOOL (Wang et al., 2019a), ATTNPOOL (Girdhar & Ramanan, 2017) as well as second order pooling methods $SOPool_{bimap}$ and $SOPool_{attention}$ (Wang & Ji, 2020).

## 5 RESULTS & DISCUSSION

The results of our experiments are summarized in Table 3 and Table 4 .From the results we can see that our methods lead to an improvement in classification accuracy over existing methods and

Table 3: Comparison results of our proposed methods with other graph pooling methods on bioinformatics datasets. Results shown are the average classification accuracy and standard deviation across 10-fold cross-validation.

| | PTC | PROTEINS | MUTAG | NCI1 |
|---|---|---|---|---|
| SUM/AVG(Xu et al., 2019) | $64.6 \pm 7.0$ | $76.2 \pm 2.8$ | $89.4 \pm 5.6$ | $82.7 \pm 1.7$ |
| DIFFPOOL(Ying et al., 2018) | $66.1 \pm 7.7$ | $78.8 \pm 3.1$ | $94.8 \pm 4.8$ | $76.6 \pm 1.3$ |
| SORTPOOL(Zhang et al., 2018) | $69.5 \pm 6.3$ | $79.2 \pm 3.0$ | $95.2 \pm 3.9$ | $78.9 \pm 2.7$ |
| TOPKPOOL(Gao & Ji, 2019) | $68.4 \pm 6.4$ | $79.1 \pm 2.2$ | $94.7 \pm 3.5$ | $79.6 \pm 1.7$ |
| SAGPOOL(Lee et al., 2019) | $69.0 \pm 6.6$ | $78.4 \pm 3.1$ | $93.9 \pm 3.3$ | $79.0 \pm 2.8$ |
| ATTNPOOL(Girdhar & Ramanan, 2017) | $71.2 \pm 8.0$ | $77.5 \pm 3.3$ | $93.2 \pm 5.8$ | $80.6 \pm 2.1$ |
| EIGENPOOL(Ma et al., 2019b) | - | $76.6 \pm 2.3$ | $80.6 \pm 4.3$ | $77.0 \pm 2.3$ |
| COVPOOL(Wang et al., 2019a) | $73.3 \pm 5.1$ | $80.1 \pm 2.2$ | $95.3 \pm 3.7$ | $83.5 \pm 1.9$ |
| SOPOOL$_{attn}$(Wang & Ji, 2020) | $72.9 \pm 6.2$ | $79.4 \pm 3.2$ | $93.6 \pm 4.1$ | $82.8 \pm 1.4$ |
| SOPOOL$_{bimap}$(Wang & Ji, 2020) | $75.0 \pm 4.3$ | $80.1 \pm 2.7$ | $95.3 \pm 4.4$ | $83.6 \pm 1.4$ |
| Neural Pooling 1(ours) | $74.5 \pm \mathbf{3.7}$ | $\mathbf{80.6} \pm \mathbf{2.7}$ | $94.0 \pm \mathbf{2.3}$ | $83.1 \pm \mathbf{1.2}$ |
| Neural Pooling 2(ours) | $\mathbf{76.2} \pm \mathbf{4.2}$ | $79.6 \pm 3.0$ | $\mathbf{95.5} \pm \mathbf{2.4}$ | $83.4 \pm 1.9$ |

Table 4: Comparison results of our proposed methods with other graph pooling methods on social network datsets. Results shown are the average classification accuracy and standard deviation across 10-fold cross-validation.

| | COLLAB | RDT-B | IMDB-B | IMDB-M | RDT-M5K |
|---|---|---|---|---|---|
| SUM/AVG | $80.2 \pm 1.9$ | $92.4 \pm 2.5$ | $75.1 \pm 5.1$ | $52.3 \pm 2.8$ | $57.5 \pm 1.5$ |
| DIFFPOOL | $75.3 \pm 2.2$ | - | $74.4 \pm 4.0$ | $50.1 \pm 3.2$ | - |
| SORTPOOL | $78.2 \pm 1.6$ | $81.6 \pm 4.6$ | $77.5 \pm 2.7$ | $53.1 \pm 2.9$ | $48.4 \pm 4.8$ |
| TOPKPOOL | $79.6 \pm 2.1$ | - | $77.8 \pm 5.1$ | $53.7 \pm 2.8$ | - |
| SAGPOOL | $78.9 \pm 1.7$ | - | $77.8 \pm 2.9$ | $53.1 \pm 2.8$ | - |
| ATTNPOOL | $81.8 \pm 2.2$ | $92.5 \pm 2.3$ | $77.1 \pm 4.4$ | $53.8 \pm 2.5$ | $57.9 \pm 1.7$ |
| COVPOOL | $79.3 \pm 1.8$ | $90.3 \pm 3.6$ | $72.1 \pm 5.1$ | $47.8 \pm 2.7$ | $58.4 \pm 1.7$ |
| SOPOOL$_{attn}$ | $81.1 \pm 1.8$ | $91.7 \pm 2.7$ | $78.1 \pm 4.0$ | $54.3 \pm 2.6$ | $58.3 \pm 1.4$ |
| SOPOOL$_{bimap}$ | $79.9 \pm 1.9$ | $89.6 \pm 3.3$ | $78.4 \pm 4.7$ | $54.6 \pm 3.6$ | $58.4 \pm 1.6$ |
| Neural Pooling 1(ours) | $80.5 \pm \mathbf{1.5}$ | $90.6 \pm \mathbf{2.3}$ | $\mathbf{79.0} \pm \mathbf{2.3}$ | $\mathbf{55.1} \pm \mathbf{2.2}$ | $\mathbf{58.5} \pm 1.8$ |
| Neural Pooling 2(ours) | $81.0 \pm 1.7$ | $91.5 \pm 3.0$ | $\mathbf{78.5} \pm \mathbf{2.4}$ | $54.4 \pm \mathbf{1.9}$ | $\mathbf{59.1} \pm \mathbf{1.4}$ |

are also more reliable as compared previous works as observed from the lower values of standard deviation.This enhancement in performance is consistent across all the datasets. The results may be attributed to the fact that compared to existing graph pooling methods, our pooling methods are able to use information from all nodes, collect second-order statistics, and leverage the ability of neural networks to learn from underlying data, making them more powerful. The Neural Pooling methods utilize the ability of neural networks to learn the topological structure as well as correlation among the node representations in, capturing essential features and connections between underlying data.

## 6 COMPLEXITY

Consider a graph $\mathcal{G} = (\boldsymbol{A}, \boldsymbol{X})$ represented by its adjacency matrix $\boldsymbol{A} \in \{0,1\}^{n \times n}$ and node feature matrix $\boldsymbol{X} \in \mathbb{R}^{n \times d}$, where $n$ is the number of nodes in $\mathcal{G}$ and $d$ is the dimension of node features. Consider, $\boldsymbol{H} = [\boldsymbol{h}_1, \boldsymbol{h}_2, ........, \boldsymbol{h}_n]^T = \text{GNN}(\boldsymbol{A}, \boldsymbol{X}) \in \mathbb{R}^{n \times f}$ where rows of $\boldsymbol{H}$, $\boldsymbol{h}_i \in \mathbb{R}^f$, $i = 1, 2, ..., n$, are representations of $n$ nodes. Consider a direct application of second-order graph pooling to obtain the graph representation $\boldsymbol{h}_G$ as:

$$\boldsymbol{h}_G = \text{Flatten}(\boldsymbol{H}^T \boldsymbol{H}) \in \mathbb{R}^{f^2 \times 1} \tag{11}$$

where $\boldsymbol{H}^T$ denotes the transpose of $\boldsymbol{H}$.

However, it causes an explosion in the number of training parameters in the following classifier when $f$ is large, making the learning process harder to converge and easier to overfit. While each layer in a GNN usually has outputs with a small number of hidden units (e.g. 16, 32, 64), it has been pointed out that graph representation learning benefits from using information from outputs of all layers, obtaining better performance and generalization ability. It is usually achieved by concatenating outputs across all layers in a GNN. In this case, $\boldsymbol{H}$ has a large final $f$, making direct use of second-order pooling infeasible. For example, if a GNN has 5 layers and each layer's outputs have 32 hidden units, $f$ becomes $32 \times 5 = 160$. Suppose $\boldsymbol{h}_G$ is sent into a 1-layer fully-connected classifier for $c$ graph categories in a graph classification task. It results in $160^2 c = 25,600c$ training parameters, which is excessive. We omit the bias term for simplicity. On the other hand, both of our proposed novel graph pooling methods significantly reduce the number of training parameters. In the case of Neural Pooling Method 1, considering the previous example if $f'$ is chosen to be 64, and $f$ is 160, then the total number of trainable parameters in the 2 $FCLs$ and a 1-layer fully-connected $c$ class classifier will be $(160 \times 64) + 64 + 64c = 10,304 + 64c$ notably reducing the number of parameters as compared to $25,600c$. In the case of Neural Pooling Method 2, if $f''$ is chosen to be 64, $f'$ as 32 and $f$ is 160, then the total number of trainable parameters in the 2 $FCLs$ and a 1-layer fully-connected $c$ class classifier will be $(160 \times 64) + (64 \times 32) + 322c = 12,288 + 1024c$ again reducing the number of parameters when compared to $25,600c$.

## 7 CONCLUSION

In this work, we propose to perform graph representation learning with Neural Pooling, by pointing out that Neural Pooling can naturally solve the challenges of graph pooling. Neural Pooling is more powerful than existing graph pooling methods, since it is capable of using all node information, collecting second-order statistics that encode feature correlations and topology information and leverage the ability of neural networks to learn from underlying data, making them more powerful. Our proposed methods solve the practical problems incurred by directly using second-order pooling with GNNs. To demonstrate the effectiveness and superiority of our methods, we perform experiments on graph classification tasks in the bio-informatics and social network domains to demonstrate the effectiveness and superiority of our proposed methods. Experimental results show that our methods improve the performance significantly and consistently. An interesting future work direction could be to extend our methods for hierarchical graph pooling, where the output is a is a pseudo graph with fewer nodes than the input graph. It is used to build hierarchical GNNs, where hierarchical graph pooling is used several times between GNN layers to gradually decrease the number of nodes.

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
