# OpenReview forum: "Neural Pooling for Graph Neural Networks"
_ICLR.cc/2021/Conference — Reject_

### Official Review · AnonReviewer4 · 2020-10-25
**AnonReviewer4**

**Rating:** 3
**Confidence:** 4

**Review:**

This paper proposes two fully-connected layers based neural graph pooling methods for graph neural networks, named Neural Pooling Method 1 and Neural Pooling Method 2. The first method uses a first FC to reduce the feature dimension and then FC2 to compute the weights to do weighted-average over features for different nodes. The second method uses two FC to reduce the dimension and then compute second-order statistics by Flatten(H^{\top}H). Experimental results on four datasets (PTC, PROTEINS, IMDB-BINARY, IMDB-MULTI) of two tasks (bioinformatics, social networks) show that the proposed graph pooling method can improve the performance by 0.5%-1.2% accuracy while decreasing the std.

Strengths:
- The proposed method is simple and motivated by several limitations of current graph pooling methods such as average and summation, DIFFPOOL, SORTPOOL, TOPKPOOL, SAGPOOL, and EIGENPOOL.

- The proposed approach is simple and the experimental results can deliver improvements on several tasks and datasets.

Weaknesses:
- My biggest concern is that the proposed approach lacks originality and novelty because it is a simplification and variant of SOPOOL from Second-Order Pooling for Graph Neural Networks (Ji and Wang, 2020)

- Based on the author's writing, it is unclear what is the second-order statistics for graph pooling, why it is important to have second-order pooling, and how the proposed method can capture the second-order statistics.

- The proposed graph pooling method is only experimented with 1 underlying particular choice of GNN (Xu et al., 2019), so it is unclear how well the method can perform on other GNN architectures.

- The four datasets only have 2 or 3 classes and upto 620 nodes. So it is clear how well the method can generalize to large-scale graph classification problems.

- The improvement of the proposed methods compared with SOPpool is marginal. For example, On PROTEINS, the accuracy is improved by 0.5% with the same std. On other datasets, the improvements are only at most 1.2%. To show the proposed approach is better, more datasets or tasks should be used. For example, there are five bioinformatics datasets (MUTAG, PTC, PROTEINS, NCI1, DD) and five social network datasets (COLLAB, IMDB-BINARY, IMDB-MULTI, REDDIT-BINARY, REDDIT-MULTI5K).

- There is not enough discussion and analysis of the results. Especially, there should be some analysis to compare the method 1 and method 2: For different datasets, when one method is better than the other? Some examples would be helpful, too.

- While the author explains the proposed method has lower complexity, there is still no formal analysis or quantitative measures of running time from experiments.

- The writing can be improved, In the abstract and introduction, the author should describe the approach briefly and explain its characteristics including why it can handle variable number of nodes, invariant to isomorphic graph structures, capture information of all nodes, and especially why it can collect second-order statistics.

- Furthermore, there is a lot of repetition of problem statements. The problem and notation is introduced formally in section 3.1, but is repeated again and again at the beginning of section 3.2 and section 3.3

Questions:
- Do both of your method 1 and method 2 capture second-order statistics? My understanding is that only method 2 captures second-order statistics by computing Flatten(H^{\top}H). Is this correct?
- How do you compare your method with SOPpool (Ji and Wang, 2020)?
- Have you tried other datasets or other tasks?
- Have you tried your graph pooling approaches on other underlying GNN models?
- Is your standard deviation in Table 2 based on 1 run of 10 folds or multiple runs of 10-fold cross-validation?

Minor:
- Please give better names for your approaches and give a better title. "Neural Pooling Method" is too general and thus not particular enough to summarize your method.

---

> ### Author Response · Authors · 2020-11-24
> **Response to Reviewer 4**
>
> Dear Reviewer2, thank you for your valuable comments. We attempt to address your concerns as follows:
>
> 1)Do both methods capture second order statistics:Yes, only method 2 captures second-order statistics by computing Flatten(HTH).
>
> 2)In this work we propose two novel techniques for graph pooling Neural Pooling Method 1 and Method 2. In Z. Wang and S. Ji. Second-order pooling for graph neural networks. IEEE Transactions on Pattern Analysis and Machine Intelligence, 2020 they propose 2 techniques SOPoolBimap and SOPoolAttention. SOPool Bimap employs a single linear mapping on node features H to perform dimensionality reduction. SOPoolAttention uses a single 1D trainable vector to reduce dimensionality. In our work we use two fully connected neural networks. The first fully connected neural network performs the role of reducing the dimensionality of the input node representations. The trainable parameters of this layer can be thought of as learning a mapping from a higher to a lower dimensional space. The second layer reduces the node representations to a 1 dimensional representation, Q. Q can be thought of as roughly encoding the position of nodes by learning the weights according to which a particular feature is aggregated across the nodes.
>
> 3)We have also included the performance of our methods on 5 more datasets, making a total of 4 bioinformatics and 5 social network datasets.
>
> 4)The standard deviation in Table 2 is based on 1 run of 10 folds of 10-fold cross-validation.
>
> 5)We have made the corrections in the diagrams to resolve some issues and improve their legibility . We have removed the redundant occurrences of paragraphs and equations. We have added appropriate punctuation wherever necessary. We have made changes to avoid the use of long sentences to improve readability.

---

### Official Review · AnonReviewer2 · 2020-10-28
**The submission proposed to use “permutation invariance property” to constrain the pooling of node embeddings to obtain the whole graph embedding for downstream tasks. However, the paper is with very low quality and hard to read. The contribution seems limited.**

**Rating:** 2
**Confidence:** 5

**Review:**

1). The claim of “the variable number of node representations as inputs and producing fixed-sized graph representations” is weak. This can be easily addressed by simply using average pooling or sum pooling.



2).. The writing needs to be improved. The quality is very low. For example, even in the introduction, where is the ending of the second paragraph? some  full stop mark is needed. “Neural Pooling Method 1 and 2”, it is better to give them specific names for better reference. “After this H0 is again passed through”, simply not readable..

3). “Neural Pooling Method 2” lacks of intuition. Could you please provide more intuitions on the solution during the rebuttal?


4). The results reported seems much worse than results reported in other paper. For example, in the paper below. We can easily see the best result in PTC, best result is 80.41±6.92 while the submission gives only 76.2 ± 4.2  From that perspective, I did not see any advantage in the submission.

Structural Landmarking and Interaction Modelling: on Resolution Dilemmas in Graph Classification,
https://arxiv.org/pdf/2006.15763.pdf

5). The paper is lack of parameter sensertivity analysis, without which the robustness of the proposed algorithms is unknown to me. I would suggest the authors add additional section to discuss that.

6). The draft includes an github link to share the code, however that link indicates the author affiliation information. Clearly I could not access the code on the github as I would have risked infringing anonymity.

---

> ### Author Response · Authors · 2020-11-24
> **Response to Reviewer 2**
>
> Dear Reviewer2, thank you for your valuable comments. We attempt to address your concerns as follows:
>
> 1)We have added the comparison results with EigenPool in the revised manuscript. We have also included the performance of our methods on 5 more datasets, making a total of 4 bioinformatics and 5 social network datasets.
>
> 2)We have made the corrections in the diagrams to resolve some issues and improve their legibility . We have removed the redundant occurrences of paragraphs and equations. We have added appropriate punctuation wherever necessary. We have made changes to avoid the use of long sentences to improve readability.
>
> 3)Intuition for Method 2: Tapping the node representation H after FCL2 gives cascade compression of the information and introduction of nonlinearity.

---

### Official Review · AnonReviewer1 · 2020-10-29
**Official Blind Review #1**

**Rating:** 4
**Confidence:** 5

**Review:**


In this paper, the authors proposed two graph pooling methods, i.e., Neural Pooling Method 1 and 2. Both of them are flat pooling strategies, which try to obtain a graph representation directly from its node representations without coarsening graphs step by step. Specifically, the major idea of Neural Pooling Method 1 is to use GCN layer to learn a score for each node. Then, the graph representation is obtained by weighted summing the node representations with the learned scores as weights.  Neural Pooling Method 2 follows a similar design. The difference is that, instead of a single score, it has multiple scores for each node, which leads to a matrix for graph representation. This matrix is then flattened into a vector to serve as the graph representation.

In general, the novelty of this paper is limited. Some other concerns are listed as follows:
It is not clearly motivated why the topology information can be preserved by the two proposed pooling method. It would be better if the authors could provide more explanation.
The process in Equation (6) can be viewed as a weighted summation. However, the values in $Q$ seem to be unbounded, which makes the magnitude of the graph representation h_G highly dependent on the size of graphs (i.e., number of nodes). Is it designed in this way on purpose to capture the node size information?  The same issue exists in the Neural Pooling Method 2.
It would be better if the users could adopt more datasets should for experiments.
Minor comments:
When analyzing the complexity of algorithms, it might be better to use general notations instead of concrete numbers.

---

> ### Author Response · Authors · 2020-11-24
> **Response to Reviewer 1**
>
> Dear Reviewer1, thank you for your valuable comments. We attempt to address your concerns as follows:
>
> 1)How is topology information preserved:\
>  In our work we use two fully connected neural networks. The first fully connected neural network performs the role of reducing the dimensionality of the input node representations. The trainable parameters of this layer can be thought of as learning a mapping from a higher to a lower dimensional space. The second layer reduces the node representations to a 1 dimensional representation, Q. Q can be thought of as roughly encoding the position of nodes by learning the weights according to which a particular feature is aggregated across the nodes, hence preserving topology information
>
> 2)Regarding weighted summation: The process in Equation (6) can be viewed as a weighted summation. Yes it is  designed in this way on purpose to capture the node size information.
>
> 3)Regarding datasets: We have now included the performance of our methods on 5 more datasets in the revised manuscript, making a total of 4 bioinformatics and 5 social network datasets.

---

### Official Review · AnonReviewer3 · 2020-10-29
**Novelty is limited; Insufficient datasets to demonstrate the results.**

**Rating:** 3
**Confidence:** 5

**Review:**

In this manuscript, the authors propose two novel methods using fully connected neural network layers for graph pooling, namely Neural Pooling Method 1 and 2. compared to existing graph pooling methods, the authors think their methods are able to capture information from all nodes, collect second-order statistics, and leverage the ability of neural networks to learn relationships among node representations, making them more powerful.

Pros :
1. This work studies an important topic but less explored topic, graph pooling.
2. Propose to perform graph representation learning with Neural Pooling.
3. Experimental results are interesting.

Cons:
1. The main concern is the lack of novelty, and the technical contribution is very limited. The essential difference between the author’s manuscript and this article is unclear -- ‘Z. Wang and S. Ji. Second-order pooling for graph neural networks. IEEE Transactions on Pattern Analysis and Machine Intelligence, 2020.’? Just one more layer? And there is a lot of overlap in content with this article.

2. It is actually a stealth exchange of concepts that the authors attribute the success of the methods to the neural networks. On the one hand, the reason $H^{'^T}Q$ or  $H^{'^T}H^{'}$ can capture topology information is not that the neural networks can learn $H'$ that contains topology information, but that $H$ itself contains local topology information and $H^{T}H$ is capable of capturing second-order statistics. In fact, the neural networks don’t use topological structure. They play a vital role in reducing dimension and parameters, which is proven in Section 6. On the other hand, $Q$ can be thought of as the weight of the node, not as the correlation among the node representations. Therefore, the statement ‘Neural networks can learn the correlation among the node representation’ lack of further explanation.

3. About the experiment of this manuscript:
a.	Several advanced pooling methods are ignored, especially EigenPooling. Why does it appear in the method comparison in Section 4.3, but disappear in Table 2.
b.	Datasets are not enough. Just 4 of the 9 data sets in this article are used -- ‘Z. Wang and S. Ji. Second-order pooling for graph neural networks. IEEE Transactions on Pattern Analysis and Machine Intelligence, 2020.’

4. About the clarity of this manuscript:
a.	Figures 1 and 2 are not beautiful and have minor issues. $H^{'^T}Q$ and $QH^{'^T}$ are not equal, which can be corrected by slightly changing the diagrams.
b.	Repeat a paragraph and an equation many times. For example, in Sections 3.1, 3.2, 3.3 and Section 6, there are redundant.
c.	The lack of punctuation, such as the end of the second paragraph of Section 1 on the first page, and the end of the second last paragraph of Section 3.3 on page 5, etc.
d.	Many long sentences make understanding difficult. For example, in section 3.2, there is only one long sentence in a paragraph.

---

> ### Author Response · Authors · 2020-11-24
> **Response to Reviewer 3**
>
> Dear Reviewer3, thank you for your valuable comments. We attempt to address your concerns as follows:
>
> 1)Novelty: In this work we propose two novel techniques for graph pooling Neural Pooling Method 1 and Method 2. In Z. Wang and S. Ji. Second-order pooling for graph neural networks. IEEE Transactions on Pattern Analysis and Machine Intelligence, 2020 they propose 2 techniques SOPoolBimap and SOPoolAttention. SOPool Bimap employs a single linear mapping on node features H to perform dimensionality reduction. SOPoolAttention uses a single 1D trainable vector to reduce dimensionality. In our work we use two fully connected neural networks. The first fully connected neural network performs the role of reducing the dimensionality of the input node representations. The trainable parameters of this layer can be thought of as learning a mapping from a higher to a lower dimensional space. The second layer reduces the node representations to a 1 dimensional representation, Q. Q can be thought of as roughly encoding the position of nodes by learning the weights according to which a particular feature is aggregated across the nodes.
>
> 2)Correlation among node representations: The node representations H ∈ R n × f can be viewed as H = [l1, l2, . . . , lf ]. The vector lj encodes the spatial distribution of the j-th feature in the graph. Based on this view, HTQ is able to capture the topology information and Q can be thought of as capturing the relations among node representations by learning the weights according to which the j-th feature is aggregated across the nodes.
>
> 3)Regarding the  experiment:\
> --We have added the comparison results with EigenPool in the revised manuscript.\
> --We have also included the performance of our methods on 5 more datasets, making a total of 4 bioinformatics and 5 social network datasets.
>
> 4)Regarding clarity of manuscript:\
> --We have made the corrections in the diagrams to resolve the issues that were mentioned.\
> --We have removed the redundant occurrences of paragraphs and equations.\
> --We have added appropriate punctuation wherever necessary.\
> --We have made changes to avoid the use of long sentences to improve readability.

---

### Decision · Program_Chairs · 2021-01-07
**Final Decision**

**Decision:**

Reject

**Comment:**

All four knowledgeable referees have indicated reject due to many concerns. In particular, reviewers pointed out that the novelty of this paper is not clear because the difference from related work is very limited (i.e., the difference from Z. Wang and S. Ji is not clear, other than using one additional layer),  and they were concerned that the results of the experiment are not convincing (For example, the results reported in this paper are significantly inferior to those reported in other papers, the GNN architecture used is limited, and the performance difference especially in the additional experiments in the revision, is very marginal). No reviewers were convinced by the authors' claims even through the author's rebuttal and revision.

One important note: Reviewers have stated that they did not explicitly check the identity of the author and did not pose a problem on this, but if we follow the link specified in the original submission, we can see the identity of the author, which may be considered as a violation of the double-blind policy. This is a small and regrettable mistake, but it can be a serious problem in the review process. In this review process, reviewers unanimously suggested rejection even ignoring this issue, but it seems that you need to pay attention in your future submissions.